# P2X7 Receptor Modulation of the Gut Microbiota and the Inflammasome Determines the Severity of *Toxoplasma gondii*-Induced Ileitis

**DOI:** 10.3390/biomedicines11020555

**Published:** 2023-02-14

**Authors:** Aline Cristina Abreu Moreira-Souza, Hayandra Ferreira Nanini, Thuany Prado Rangel, Sthefani Rodrigues Batista da Silva, Beatriz Pêgo Damasceno, Beatriz Elias Ribeiro, Cynthia M. Cascabulho, Fabiano Thompson, Camille Leal, Patrícia Teixeira Santana, Siane Lopes Bittencourt Rosas, Kívia Queiroz de Andrade, Claudia L. Martins Silva, Rossiane Claudia Vommaro, Heitor Siffert Pereira de Souza, Robson Coutinho-Silva

**Affiliations:** 1Laboratório de Imunofisiologia, Instituto de Biofísica Carlos Chagas Filho, Universidade Federal do Rio de Janeiro, Rio de Janeiro 21941-590, Brazil; 2Departamento de Clínica Médica, Hospital Universitário Clementino Fraga Filho, Universidade Federal do Rio de Janeiro, Rio de Janeiro 21941-913, Brazil; 3Laboratório de Inovações em Terapias, Ensino e Bioprodutos, Instituto Oswaldo Cruz, Fundação Oswaldo Cruz, Rio de Janeiro 21040-360, Brazil; 4Instituto de Biologia, Universidade Federal do Rio de Janeiro, Rio de Janeiro 21941-901, Brazil; 5Laboratório de Farmacologia Bioquímica e Molecular, Instituto de Ciências Biomédicas, Universidade Federal do Rio de Janeiro, Rio de Janeiro 21941-902, Brazil; 6Laboratório de Ultraestrutura Celular Hertha Meyer, Instituto de Biofísica Carlos Chagas Filho, Universidade Federal do Rio de Janeiro, Rio de Janeiro 21941-170, Brazil; 7Instituto D’Or de Pesquisa e Ensino, Rio de Janeiro 22281-100, Brazil

**Keywords:** gut microbiota, inflammasome, purinergic signaling, P2X7 receptor, *Toxoplasma gondii* ileitis

## Abstract

In mice, oral *Toxoplasma gondii* infection induces severe ileitis. The aim of the present study was to investigate the impact of the P2X7 receptor (P2X7) on the inflammatory response to *T. gondii*-induced ileitis. Cysts of the ME49 strain of *T. gondii* were used to induce ileitis. The infected mice were euthanized on day 8 and ileal tissue and peripheral blood were collected for histopathological and immunohistochemical analyses. Ileal contractility, inflammatory mediators, inflammasome activation, quantitative PCR analysis of gene expression, and fecal microbiota were assessed using appropriate techniques, respectively. The infected P2X7−/− mice had greater disease severity, parasitic burden, liver damage, and intestinal contractility than the infected wild-type (WT) mice. Infection increased serum IL-6 and IFN-γ and tissue caspase-1 but not NLRP3 in P2X7−/− mice compared to WT mice. Bacteroidaceae, Rikenellaceae, and Rhodospirillales increased while Muribaculaceae and Lactobacillaceae decreased in the infected WT and P2X7−/− mice. Bacteroidia and Tannerellaceae increased in the P2X7−/− mice with ileitis. By contrast, Clostridiales and Mollicutes were absent in the P2X7−/− mice but increased in the WT mice. P2X7 protects mice against *T. gondii* infection by activating the inflammasome and regulating the local and systemic immune responses. Specific gut bacterial populations modulated by P2X7 determine disease severity.

## 1. Introduction

*Toxoplasma gondii* is an intracellular parasite in Phylum Apicomplexa. It causes one of the most common human parasitic infections and has been extensively utilized in murine infection models [1]. Although chronic *T. gondii* infection is estimated to involve almost one-third of the human population, most infections are asymptomatic in healthy hosts but can become severe in immunocompromised individuals [2]. After oral ingestion, the walls of the *T. gondii* cyst disrupt in the host stomach, tachyzoites are released, and they disseminate through the distal ileum, where they induce inflammation [3]. In mice, oral *T. gondii* infection causes ileitis, which resembles human inflammatory bowel disease (IBD) and has been proposed as an experimental model of Crohn’s disease ileitis [4]. In mice orally exposed to *T. gondii*, inflammatory cells rapidly accumulate and create a milieu predominated by the Th1-type immune response [5]. The immune and inflammatory responses against *T. gondii* infection are vital for the control of parasite replication and proliferation within host tissues. Nevertheless, the production of proinflammatory mediators can also lead to tissue damage and is pivotal in disease pathogenesis.

Tissue injury always releases adenosine triphosphate (ATP), which activates purinergic receptors. The P2X7 receptor (P2X7) has been extensively studied and occurs in epithelial and immune cells [6]. The ATP-P2X7 pathway modulates cytokine production, inflammatory mediator secretion, inflammasome activation, and other intracellular signaling pathways. Hence, it participates in the pathogenesis of several inflammatory diseases [7]. P2X7 expression in immune cells has been implicated in the control of *T. gondii* and other intracellular infections in humans. P2X7 mutations are associated with augmented susceptibility to *T. gondii* [8]. In vivo, *T. gondii* infection rapidly upregulates epithelial C-C motif chemokine ligand 5 (CCL5) and recruits dendritic cells to the epithelium in WT but not P2X7−/− mice [9]. In another in vivo study, *T. gondii* infection in human fetal small intestinal epithelial cells revealed the roles of the P2X7/NLRP3 pathway in promoting IL-1β secretion and inhibiting *T. gondii* proliferation [10]. A study on peritoneal and bone marrow-derived macrophages showed that the control of *T. gondii* via P2X7 depends on the canonical NLRP3 inflammasome. The latter increases IL-1β production via caspase-1 activity, induces mitochondrial reactive oxygen species generation, and eliminates parasites [11]. The role of the P2X7 in *T. gondii* infection-induced intestinal disease has been previously investigated. Nevertheless, several questions about this process remain unanswered.

The precise molecular mechanisms involved in toxoplasmosis-induced ileitis are not fully understood. Nonetheless, there is evidence to suggest that the gut microbiota triggers the inflammatory process through lipopolysaccharide (LPS)-induced TLR-4 signaling [12]. Another experimental study identified changes in the gut microbiota that were associated with the development of *T. gondii* ileitis [13]. In addition, the indigenous gut microbiota was shown to protect the host against certain infections [14,15]. Thus, we hypothesized that compositional changes in the gut microbiome may precede the onset of *T. gondii*-induced ileitis. In the present study, we used an in vivo experimental mouse *P2X7R* knockout model to clarify the potential roles of the P2X7 receptor in regulating the events promoting *T. gondii*-associated ileitis and influencing the gut microbiota.

## 2. Materials and Methods

### 2.1. Animal Study Ethics Statement

The Institutional Animal Care Committee of the Health Sciences Centre of the Federal University of Rio de Janeiro approved the care and use of the animals and procedures reported in the present study (approval ID: 086/18) in accordance with the guidelines of Animal Research: Reporting of In Vivo Experiments (ARRIVE) (www.nc3rs.org.uk/ARRIVE (accessed on 10 January 2022)) developed by the National Centre for the Replacement, Refinement, and Reduction of Animals in Research (NC3Rs).

### 2.2. Animals

All mice in each experimental group of this study were housed in the same rack-mounted wire cage. It had sufficient space to accommodate all mice and reduce the heterogeneity in gut microbiota composition between individuals [16]. Standard laboratory pelleted formula and autoclaved water were provided ad libitum. Female C57BL/6 wild type (WT) and P2X7 knockout (P2X7−/−) mice aged 6–8 weeks were procured from The Jackson Laboratory (Bar Harbor, ME, USA) and maintained at 20–24 °C under a 12 h/12 h light-dark cycle and specific pathogen-free conditions. Carworth Farm mice (CF-1) were obtained from the State University of Campinas (São Paulo, Brazil) and used to maintain *Toxoplasma gondii* passage. The CF-1 mice were orally infected by gavage with a maximum volume of 100 mL phosphate-buffered saline (PBS) containing 50 brain cysts of nonvirulent type II *T. gondii* ME-49 strain. The WT and P2X7−/− mice were orally infected by gavage with ten ME-49 cysts. Orally infected mice were monitored once weekly for one month after infection. Susceptibility to infection was determined based on mouse survival.

### 2.3. Maintenance of T. gondii Cysts

*T. gondii* was maintained using Swiss mice aged 6–8 weeks. Each mouse was orally infected by gavage with 10 ME-49 cysts. After two months, the mice were euthanized in a CO_2_ chamber. Death was confirmed by cervical dislocation. The brains were excised and macerated in PBS. The cysts in the brain homogenates were counted and used either to infect other mice and maintain the parasite reservoir or to induce ileitis.

### 2.4. Experimental Design

After an acclimation period of one week, the WT and P2X7−/− mice were randomly allocated to the experimental groups and orally infected by gavage with 10 ME-49 cysts. Body weight was measured daily and used as a clinical parameter of ileitis development. On day 8 of infection, the mice were euthanized in a CO_2_ chamber. Blood samples were collected by cardiac puncture and the small intestines were excised, measured, weighed, and washed with saline. Ileal fragments were collected and processed for the subsequent analyses. The mice were enumerated with G*Power (https://g-power.apponic.com (accessed on 8 February 2021)) and two-way ANOVA. The input settings were Cohen’s F = 0.3, type I error = 0.05, power = 0.8, and numerator df = 80.

### 2.5. Ileal Parasite Load

Details of parasite load determination are described in the Appendix A.

### 2.6. Aminotransferase Assays

The blood was centrifuged at 9000× *g* for 5 min. Then 20 μL serum was added to each well of a 96-well plate. Each well contained 100 μL Master Reaction Mix (Bioclin, Belo Horizonte, Brazil) consisting of enzyme mix, developer, substrate, and assay buffer. The plates were shielded from light, mixed, and incubated at 37 °C for 5 min. Absorbance was measured at 340 nm in a SpectraMax plate reader to detect aspartate transaminase (AST) and alanine transaminase (ALT) according to the manufacturer’s protocol.

### 2.7. Cytokine Measurements

The blood was centrifuged at 9000× *g* for 5 min. The plasma was collected and its cytokine content was measured with a Cytometric Bead Array Mouse Th1/Th2/Th17 Cytokine Kit (BD Biosciences, San Jose, CA, USA) in a FACSCalibur flow cytometer (BD Biosciences). The results were generated and analyzed with BD CBA Analysis software (BD Biosciences). The total protein content of the biopsy specimens was estimated with a Pierce bicinchoninic acid (BCA) protein assay kit (Thermo Fisher Scientific, Rockford, IL, USA) and used to normalize the results.

### 2.8. Histological Analysis

Ileal samples were immediately fixed in 40 g/L formaldehyde-saline for ≥24 h and embedded in paraffin. The ileal tissues were cut into 5-μm sections, stained with hematoxylin and eosin (H&E), and microscopically examined. Histological parameters including ulceration, hyperplasia, and inflammatory infiltration were evaluated to calculate the inflammatory score. Details of the histological evaluation are described in the Appendix A.

### 2.9. Immunohistochemistry (IHC) and Apoptosis Assessment

Details of the IHC experiments are described in the Appendix A.

### 2.10. Nitric Oxide (NO) Production and Myeloperoxidase Activity Assessment

Details of the NO and myeloperoxidase activity analyses are described in the Appendix A.

### 2.11. Ileal IL-1β Measurement

Ileal samples were collected, macerated in PBS on ice, and centrifuged at 400× *g* for 5 min. The supernatants were collected and used to measure the ileal IL-1β concentrations. The total protein content of the ileal fragments was quantified with a Pierce BCA protein assay kit (Thermo Fisher Scientific) and used to normalize the results. The ileal IL-1β levels were quantitated using a commercially available sensitive enzyme-linked immunosorbent assay (ELISA) method.

### 2.12. RNA Isolation, cDNA Synthesis, and qRT-PCR

RNA was extracted from the ileal fragments with TRIzol reagent (Thermo Fisher Scientific, Wilmington, DE, USA). The cDNA was constructed with a high-capacity kit (Life Technologies, Carlsbad, CA, USA) according to the manufacturer’s instructions. RT-qPCR was performed on 1 µg cDNA. Gene expression levels were validated by qRT-PCR. RT-PCR was performed in an ABI Prism 7500 (Applied Biosystems, Foster City, CA, USA) with a CustomTaqMan^®^ Array Gene Signature Plate (Thermo Fisher Scientific). Further details on the qRT-PRC and the primers are described in the Appendix A.

### 2.13. Contractile Activity of Longitudinal Smooth Muscle in the Ileum

Details of the contractile activity analysis are described in the Appendix A.

### 2.14. Fecal Microbiota Analysis

Details of the fecal microbiota analysis by high-throughput 16S rRNA sequencing are described in the Appendix A.

### 2.15. Statistical Analysis

Statistical analyses were performed in Graph Pad Prism v. 9.1.2 (GraphPad Software, San Diego, CA, USA). A two-tailed *t*-test was used for pairwise comparisons. Multiple comparisons were performed by one-way ANOVA followed by the post hoc Tukey’s test. The results are means ± standard deviations (SD) or standard errors of the mean (SEM). Survival data are presented in the form of Kaplan–Meier survival curves and were analyzed by the log-rank test. All tests were two-tailed and differences between treatments were considered statistically significant at *p* < 0.05.

## 3. Results

### 3.1. The P2X7 Receptor Is Overexpressed in the Ileum Following T. gondii Infection

To investigate P2X7 expression during intestinal inflammation induced by *T. gondii*, C57BL/6 WT and P2X7−/− mice were orally infected with ten ME-49 cysts. The expression of the P2X7 receptor was analyzed by IHC in the intestinal sections on day 8 post-infection. The results showed that the P2X7 receptor was significantly upregulated in the ileal mucosae of *T. gondii*-infected mice compared to the uninfected controls (Figure 1A,B). We analyzed molecular-level P2X7 receptor expression to confirm this finding. The P2X7 mRNA level was significantly higher in the ileal samples from *T. gondii*-infected mice than it was in those from the uninfected controls (Figure 1C).

### 3.2. The P2X7 Receptor Protects the Host against T. gondii Infection

#### 3.2.1. Survival Curve

A survival curve was plotted to characterize the roles of the P2X7 receptor in *T. gondii*-induced ileitis. Susceptibility was evaluated for 30 d after oral infection with 10 ME-49 cysts. The P2X7−/− mice were relatively more susceptible to oral infection than the WT mice (Figure 2A).

#### 3.2.2. Tissue Damage and Parasite Loading

Histological analyses with H&E staining were performed on the ileum to assess the severity of ileitis induced by *T. gondii* infection (Figure 2B). The *T. gondii* infection damaged the ileal mucosa to a significantly greater extent in the P2X7−/− than in the WT mice. The injuries were concentrated in the terminal ilea (Figure 2C).

We investigated the presence of the parasite in the intestines of the infected animals to identify the reasons for exacerbated inflammation. H&E staining showed that there were more tachyzoites in the ilea of the P2X7−/− mice than there were in those of the WT mice (Figure 2D). We examined the parasite load by cell monolayer disruption. An intact LLCM-K2 cell monolayer was cultured with 200 µL WT or P2X7−/− mouse ileum lysate for one week. Rapid monolayer disruption was observed in the presence of the parasites recovered from the P2X7−/− ileum lysate (Figure 2E). We validated the parasite load assay by quantifying parasite β-actin in the ileal fragments via semiquantitative RT-PCR. There was more *T. gondii* β-actin in the ileal fragments of the P2X7−/− mice than in those of the WT mice (Figure 2F).

No other significant differences between the WT and P2X7−/− mice in terms of intestinal injury during *T. gondii* infection were noted. Severe ileal inflammation was characterized by epithelial intestinal extrusion and dense inflammatory cell infiltration, necrosis, and ulceration. The inflammatory scores for the terminal ilea did not significantly differ between the infected WT and P2X7−/− mice (Appendix A). In both types of infected mice, the numbers of Paneth cells and their granular contents were reduced (Appendix A). The proportions of differentiated goblet cells were significantly reduced in both types of infected mice (Appendix A). The increases in collagen fiber density that occurred in response to *T. gondii* infection did not significantly differ between the infected WT and P2X7−/− mice (Appendix A). We measured ki67-cell proliferation marker expression and apoptosis by TUNEL assay to evaluate cell turnover in the ileal mucosae. Ki67 expression did not significantly differ among treatment groups. Nevertheless, the rate of apoptosis was significantly higher in the *T. gondii*-infected mice than the untreated control mice but did not significantly differ between the infected WT and P2X7−/− mice (Appendix A).

### 3.3. P2X7 Modulates Morbidity and the Inflammatory Response in T. gondii Infection

#### 3.3.1. Body Weight and Liver Aminotransferases

Mouse body weight was measured for 8 d post-infection to determine the clinical consequences of *T. gondii*-induced ileitis. The uninfected mice presented with no significant variation in body weight. During *T. gondii* infection, however, the infected WT mice and the infected P2X7−/− mice showed significant body weight loss (Figure 3A). The plasma aspartate transferase (AST) and alanine transferase (ALT) levels were measured to assess liver damage and dysfunction. The infected P2X7−/− mice presented with higher plasma ALT and AST levels than the infected WT mice (Figure 3B,C).

#### 3.3.2. Systemic Inflammatory Response

Systemic proinflammatory cytokines limit *T. gondii* infection, replication, and dissemination. We used flow cytometry to measure the proinflammatory cytokine levels in the peripheral blood. IL-6, IFN-γ, TNF-α, and IL-10 were all upregulated in both groups of infected mice. However, the IL-6 and IFN-γ levels were lower in the sera of infected P2X7−/− mice than in those of the infected WT mice (Figure 3D–G).

#### 3.3.3. Mucosal Inflammatory Response and Caspase-1 Activation

Analysis of the inflammatory cell infiltrates within the ileal samples showed high CD4-positive and CD11b-positive cell densities in the lamina propria of the infected WT and P2X7−/− mice (Appendix A). Elevated caspase-1-positive cell densities were also observed in the lamina propria of the infected WT and P2X7−/− mice (Appendix A). The ileal extracts of the mice infected with *T. gondii* presented with increased myeloperoxidase activity (Appendix A).

### 3.4. Increased Ileal Contractility Is Enhanced in the Absence of the P2X7 Receptor in T. gondii Infection

As the P2X7−/− mice presented with substantial ileal injury, we investigated whether *T. gondii* infection-induced inflammation affects intestinal physiology. We found no difference between treatment groups in terms of the lengths of their small intestines (data not shown). However, the small intestine weight was higher in the infected P2X7−/− mice than the infected WT mice (Figure 4A). Acetylcholine caused a dose-dependent increase in the contraction of ileal tissue excised from infected WT and P2X7−/− mice as well as uninfected control mice (Figure 4B). However, there was a significant difference in E_max_ for ACh-induced contraction between the WT mice (1.79 ± 0.12 mN, *n* = 4) and the P2X7−/− mice (3.71 ± 0.38 mN, *n* = 3, *p* = 0.01). Hence, ACh induced stronger ileal contraction in the uninfected P2X7−/− mice than in the uninfected WT mice (Figure 4C). Moreover, the infected WT mice presented with stronger ileal contractions (2.91 ± 0.22 mN, *n* = 5) than the uninfected WT mice (1.79 ± 0.12 mN, *n* = 4). The ileum of infected P2X7−/− mice showed the highest contraction in response to ACh (4.39 ± 0.46 mN, *n* = 3, *p* = 0.009) (Figure 4C).

### 3.5. Inflammasome-Mediated Immune Response against T. gondii Infection

#### 3.5.1. RT-qPCR

The P2X7 receptor activates the inflammasome, which, in turn, secretes IL-1β. This cytokine was shown to control macrophage infection and cerebral toxoplasmosis in different models [17]. We measured the mRNA levels of the inflammasome components and the protein levels of IL-1β in the ilea of *T. gondii*-infected WT and P2X7−/− mice. The NLRP3, NLRP6, Caspase-1, and IL-1β mRNA levels were higher in the infected WT mice than in the infected P2X7−/− mice. By contrast, the AIM2 inflammasome mRNA level was significantly lower in the infected P2X7−/− mice than in the infected WT mice (Figure 5A–F).

#### 3.5.2. The P2X7 Receptor Modulates NLRP3 and IL-1β Upregulation and NO Downregulation after *T. gondii* Infection

As *T. gondii*-induced ileitis upregulated inflammasome mRNA, we measured the protein expression levels of NLRP3 and IL-1β in the ileal tissue. After *T. gondii* infection, NLRP3 expression was higher in the ileal mucosa of infected WT mice than in infected P2X7−/− mice (Figure 6A,B). The IL-1β concentrations were significantly lower in the ileal lysates of infected P2X7−/− mice than they were in those of infected WT mice (Figure 6C). On the other hand, the production of inflammatory mediators such as nitric oxide (NO) was greater in infected P2X7−/− mice than in uninfected P2X7−/− mice (Figure 6D). Therefore, *T. gondii* infection caused an unbalanced ileal inflammation response in the absence of the P2X7 receptor.

### 3.6. The P2X7 Receptor Modulates the Microbiota Associated with T. gondii-Induced Ileitis

Twenty-four ileal samples were subjected to Deblur quality control clean-up. The values (means ± SD) were as follows: control WT: 52,780 ± 1889; induced ileitis WT: 75,335 ± 2834; control P2X7−/−: 43,565 ± 935; and induced ileitis P2X7R−/−: 111,717 ± 1682. High-throughput sequencing generated 1011229 16S rRNA sequences ~85 nt in length. The reads were then segregated into 215 operational taxonomic units (OTUs). Estimators of α-diversity were calculated based on the Shannon index. Although no significant difference in α-diversity was detected among the groups, we observed a trend toward a decreased diversity in the ileitis-induced groups (Figure 7A). A principal coordinate analysis (PCoA) revealed clustering of the P2X7−/− samples (blue and green) away from the WT samples (red and orange). Ileitis induction caused unique structural changes in the fecal microbiota. Most induced ileitis samples shifted away from the control samples highlighted with colored oval forms. However, the WT controls were unevenly distributed. Hence, there was heterogeneity within that group and a more comprehensive and accurate analysis could not be performed. The fecal microbiota differed between the control and ileitis-induced mice and between the WT and P2X7−/− control mice (Figure 7B). A Venn diagram showed that most OTUs (67%) were conserved and were shared among groups. Similar superimpositions were observed for the WT mice (76%) and P2X7−/− mice (81%) and for the ileitis-induced mice (78%) and the control mice (73%) (Figure 7C). In all treatment groups, the fecal microbiota comprised mainly the phyla Bacteroidetes and Firmicutes followed by the less abundant phyla Proteobacteria and Epsilonbacteraeota, among others. No significant differences were detected among treatment groups in terms of their bacterial phyla (Figure 7D). Identification of the hierarchies within the taxonomic system enabled us to detect several distinct compositional patterns in the treatment groups. The relative microbial abundances significantly differed between the ileitis-induced and control mice and between the WT and P2X7−/− mice (Appendix A). In the WT mice, *T. gondii*-induced ileitis induced several changes in the gut microbiota, including an increased relative abundance in Rhodospirillales order (Proteobacteria phylum), Bacteroidaceae family (Bacteroidetes phylum), Gastranaerophilales order (Cyanobacteria phylum), and a reduction in Burkholderiaceae family (Proteobacteria phylum), Eggerthellaceae family (Actinobacteria phylum), and Desulfovibrionaceae family (Thermodesulfobacteriota phylum), compared with WT uninfected mice ((Appendix A). In the P2X7−/− mice, *T. gondii*-induced ileitis induced several clear changes in the gut microbiota, including an increased relative abundance in Bacteroidaceae family and Bacteroidales order (Bacteroidetes phylum), Lachnospiraceae family (Firmicutes phylum), and Gastranaerophilales order (phylum Cyanobacteria), Saccharimonadaceae family (Saccharibacteria phylum), Rikenellaceae order (Bacteroidetes phylum), Clostridiales vadin class (Firmicutes phylum), Rhodospirillales order (Proteobacteria phylum), and reductions in Burkholderiaceae family (Proteobacteria phylum), Prevotellaceae family (Bacteroidetes phylum), and Muribaculaceae family (Bacteroidetes phylum) compared with P2X7−/− uninfected mice (Appendix A). Comparing the uninfected mice, several clear changes in the indigenous gut microbiota were detected in P2X7−/− mice, including a reduction in the relative abundance in Saccharimonadaceae family (Saccharibacteria phylum), Clostridiales vadin class and Lactobacillaceae family (Firmicutes phylum), and increases in Bacteroidales order, Prevotellaceae family, and Tannerellaceae family (Bacteroidetes phylum), compared with WT mice (Appendix A). The Firmicutes:Bacteroidetes ratios did not significantly differ among treatment groups (Appendix A).

## 4. Discussion

In the present study, we investigated whether purinergic signaling via P2X7 is involved in the pathogenesis of a murine model of *T. gondii*-induced ileitis. P2X7 deficiency markedly increased the morbidity and mortality associated with oral *T. gondii* infection. We demonstrated that P2X7-deficient mice infected with *T. gondii* had a greater intestinal parasite burden, more extensive intestinal inflammation, and higher intestinal contractility than WT mice infected with *T. gondii*. The changes in the systemic and local inflammatory responses observed in the P2X7-deficient mice infected with *T. gondii* were associated with increased NO production and decreased inflammasome activation. Moreover, substantial changes in gut bacterial abundance and diversity were detected in the P2X7-deficient mice and associated with increased severity of *T. gondii* infection.

The intestinal P2X7 receptor was associated with the severity of inflammation in human IBD [18] and in experimental colitis [19,20] and ileitis [21] models. P2X7 was related to increased susceptibility to *T. gondii*-induced ileitis [22]. A previously established model of chronic *T. gondii* infection was induced by gavage administration of five ME-49 cysts over 30 d [17]. Here, however, we administered 10 ME-49 cysts and the P2X7−/− mice began to die by day 10 after infection, whereas the WT mice began to die by day 18 after infection. Miller et al. (2015) reported exacerbation of *T. gondii*-induced ileitis in P2X7−/− mice but observed no increase in their ileal or splenic parasite loads [22]. However, another study showed that intraperitoneal injection of 20 ME-49 cysts induced ileal dysbiosis and increased nitrate levels in mice [23]. By contrast, the present study showed that infected P2X7−/− mice produced relatively more nitrate than uninfected P2X7−/− mice. Thus, microbial alterations in the intestines of P2X7−/− mice might increase inflammatory mediator production. Nonetheless, we observed relatively more severe tissue damage in the infected P2X7−/− mice, in which the infection resulted in extensive small bowel inflammation, in contrast to WT-infected mice, which showed intestinal inflammation limited to the terminal ileum. One possible explanation is that the mice could not control parasite proliferation. In fact, PCR and plate assays of histopathological intestinal sections confirmed increased numbers of tachyzoites. Uncontrolled parasite loads were reported for a chronic ME-49 infection model. The investigators detected comparatively more cysts and less inflammatory infiltration in the brains of P2X7−/− than WT mice [17].

An acute *T. gondii* infection model was induced by intraperitoneal injection of the RH strain in P2X7−/− mice. The mice presented with increased susceptibility to infection and several pathological alterations such as liver damage, clotting, adherence, and increased parasitic load compared to the WT mice. Regarding the liver, *T. gondii*-infected P2X7−/− mice developed smaller granulomas, but increased parasite load per granuloma [24]. Liver damage in P2X7−/− mice was indicated by elevated aminotransferase levels. The *T. gondii* infection also caused an imbalance in inflammatory cytokine production. IL-6, IFN-γ, IL-1β, and TNF-α are involved in immune cell migration and NK cell, lymphocyte, and monocyte activation. They were upregulated in all *T. gondii*-infected mice but particularly in infected P2X7−/− mice [22]. Proinflammatory cytokines limit *T. gondii* burdens and, therefore, the severity of the infection [25]. However, the serum IL-6 and IFN-γ levels were lower in infected P2X7−/− mice than in infected WT mice in this study. An in vitro study showed that IL-6 levels decreased during pharmacological P2X7 receptor blockade in mouse colonic epithelial (CMT-93) cells subjected to *T. gondii* infection [9]. This finding corroborates our results. In an in vivo mouse model of acute *T. gondii* infection, the P2X7 receptor was responsible for increases in serum IFN-γ greater than those in splenic CD4+ IFN-γ+ cells [24]. During acute toxoplasmosis, the P2X7 receptor participates in the immune response, thereby promoting the production of inflammatory cytokines that limit parasite burdens.

The results of this study showed increased intestinal contractility in mice infected with *T. gondii*. Similarly, a previous study demonstrated that *T. gondii* mediates alterations in the enteric nervous system [26], which regulates intestinal motility [27,28]. However, the present study showed that acetylcholine mediated myogenic intestinal contraction in a dose-dependent manner. This effect was particularly pronounced in *T. gondii*-infected P2X7−/− mice. Discrepancies in the degree of intestinal contractility in the P2X7−/− mice corroborate the results of earlier reports linking P2X7 receptor expression in enteric neurons with the regulation of intestinal motility [29,30]. The mechanisms underlying the changes in intestinal contractility observed herein are unclear. Nevertheless, we hypothesize that P2X7 deficiency may reduce ATP release from epithelial and enterochromaffin cells which, in turn, lowers inhibitory junctional potentials and, by extension, potentiates the contractions induced by ACh. Under normal conditions, neuronal P2X7 receptors have relatively less control of tonic inhibition on excitatory cholinergic motility [31]. However, in the rat colitis model, the selective P2X7 antagonist A804598 induced a significant enhancement of the contractions of inflamed tissues, which seemed to depend on the NO pathway, since the nitric oxide synthase inhibition blunted the A804598 effect [31]. Here, we found increased NO levels in the ilea of infected P2X7−/− mice. Therefore, our findings are in accordance with the putative role of P2X7 in the regulation of cholinergic-mediated contractions. For this reason, ACh-induced contraction was comparatively stronger in the ilea of the infected P2X7−/− mice than it was in those of the infected WT mice.

Consistent with previous reports on oral *T. gondii* infection, our mouse model also presented with severe intestinal inflammation involving the terminal ileum. In that region, leucocytes accumulated in a predominantly Th1-type immune response, signaling pathways were activated, and several proinflammatory mediators were secreted [32,33]. We detected increases in IL-1β in the infected WT mice but not in the infected P2X7−/− mice. We also observed NLRP3 inflammasome upregulation in the infected WT mice but not in the infected P2X7−/− mice. Although the protein levels of caspase-1 were increased in both WT and P2X7−/− mice following *T. gondii* infection, the respective caspase-1 mRNA increased only in the WT group. This could be explained by the existence of additional pathways for activating caspase-1, such as the direct induction by microorganisms, for example. In a study by Franchi et al., intracellular *Salmonella* or *Listeria* were shown to activate pro-caspase-1 in macrophages, even in the context of P2X7 receptor deficiency [34]. Therefore, we speculate that available cytosolic pro-caspase-1 may be activated, at least in part, by alternative pathways, independent of P2X7. Therefore, the innate immune response against the *T. gondii* insult was upregulated and there might be a mechanistic association between P2X7 and the inflammasome pathways. In contrast to our findings, however, previous in vitro studies showed that *T. gondii* infection may activate the inflammasomes NLRP3, NLRP6, and AIM2 and, by extension, IL-1β secretion in the FHs 74 epithelial cells of the human small intestine and in human leukemia THP-1 monocytic cells whether or not the P2X7 receptor is present [35,36]. Nonetheless, it is difficult to compare and contrast the results generated by in vitro and in vivo studies with different designs and protocols. In the current study, AIM2 inflammasome expression was lower in the P2X7−/− mice than the WT mice and decreased further still after *T. gondii* infection. Taken together, the foregoing results underscore the existence of an intricate network involving P2X7 and the inflammasome pathways, activating caspase-1, and releasing mature IL-1β [37] in response to *T. gondii* infection [10,11].

Here, the inflammatory process was associated with increased oxidative stress and cell death in the intestinal epithelial layer. Similar to previous studies [25,38], we detected elevated NO levels in response to *T. gondii* infection and to a greater extent in P2X7−/− mice than WT mice. The Th1 response is essential in order to promote the production of effector molecules, such as NO that combat protozoan parasites including *T. gondii* [39]. Other investigations showed that *T. gondii* infection per se [40] and Th1-induced nitrate production [23] could induce gut dysbiosis, which may explain the observed dissimilarities in the microbiota among the treatment groups. Alterations in local mucosal immunity align with the structural and physiological damage to the small bowel described for similar experiments [41,42]. These morphophysiological changes resembled those characteristic of Crohn’s disease [43].

Loss of the structural cells composing the ileal epithelium adjacent to the mucus-secreting and Paneth cells resulting from *T. gondii* infection might favor both microbial penetration and gut dysbiosis. Disruption of the epithelial barrier might reinforce innate immunity mechanisms against *T. gondii* infection and its associated gut dysbiosis. In the current study, the gut dysbiosis observed after *T. gondii* infection was consistent with previous investigations reporting reduced microbial diversity and expansion of phylum Proteobacteria, order Rhodospirillales, and facultative anaerobes [23]. Gut dysbiosis observed in the murine *T. gondii* infection model has been attributed to Th1-induced Paneth cell loss and low antimicrobial peptide levels [44,45]. Goblet cell loss may compromise the protective mucous layer and favor bacterial adherence and penetration that occurs mechanically or results from the lack of the trefoil factor family peptides that are co-secreted with mucins [46,47].

The *T. gondii* infection-related compositional changes we detected in both WT and P2X7-deficient mice were not limited to Proteobacteria expansion. In phylum Bacteroidetes, increases in the families Bacteroidaceae and Rikenellaceae were observed in response to *T. gondii* infection. Another study reported the same finding in SAMP1/YitFc mice that had developed spontaneous ileitis [48]. On the other hand, another study using a similar *T. gondii* infection model reported that family Lactobacillaceae (phylum Firmicutes) was reduced in mice presenting with *T. gondii*-induced ileitis [13]. Moreover, *T. gondii*-infected mice exhibited reduced abundance of the family Muribaculaceae (phylum Bacteroidetes). Members of the Muribaculaceae utilize mucus-derived monosaccharides in the gut [49]. Therefore, the results of the current study and previous literature findings support the suggestion that gut dysbiosis following *T. gondii* infection may develop due to several combined mechanisms, including the inflammatory process per se, and the mucosal barrier damage, with reduction in the production of antimicrobial peptides, allowing the selection and penetration of pathogenic microorganisms.

Regarding the potential influence of the P2X7 receptor in the gut microbiota, to our knowledge, a previous study from our group was the first to show a trend toward a higher relative abundance of Cyanobacteria and Spirochaetes in fecal samples from the P2X7−/− mice. Although the findings were associated with a possible protective role against the development of colitis-associated colorectal cancer in the P2X7−/− mice, they lacked statistical significance and the analysis was limited to the phylum level [50]. In this study, the abundance of Family Burkholderiaceae (phylum Proteobacteria) was lower in P2X7-deficient than WT mice and decreased further still after *T. gondii* infection. Similar gut dysbiosis patterns were observed in a chemically induced experimental colitis model [51]. The preceding findings strongly support the occurrence of gut dysbiosis in response to *T. gondii* infection including the expansion of aggressive strains in order Rhodospirillales and families Bacteroidaceae and Rikenellaceae. Hence, the Burkholderiaceae and Muribaculaceae might confer protection against *T. gondii* infection.

Other differences in relative gut microbial abundance may account for the different degrees of severity of *T. gondii* ileitis observed in the P2X7-deficient mice. Orders Bacteroidales and Clostridiales were relatively more abundant in the P2X7-deficient mice and expanded further still after *T. gondii* infection. Similarly, another study evaluating the impact of Toll-like receptor-9 (TLR-9) on the gut microbiota after *T. gondii* infection reported increases in Bacteroidales and Clostridiales in TLR-9-deficient mice. Though TLR-9 is not required to induce *T. gondii* ileitis, it mediates inflammatory changes in the intestinal and extra-intestinal compartments [52]. Another study reported that intestinal Na+/H+ exchanger 3 (NHE3) deletion favors the expansion of proinflammatory bacterial taxa, including Bacteroidaceae, Rikenellaceae, and Tannerellaceae and contraction of the Prevotellaceae [53]. We obtained similar findings for *T. gondii*-infected, P2X7-deficient mice. In fact, a previous study suggested that commensal bacteria can act as molecular adjuvants during *T. gondii* infection. In that study, mucosal innate and adaptive immune responses to dendritic cell stimulation by normal gut microbiota conferred protection against *T. gondii* in the absence of TLR11 [54]. The foregoing findings suggest that compositional changes in the gut microbiota may precede the onset of *T. gondii*-induced ileitis and could explain the relatively greater disease severity in infected P2X7-deficient mice. Taken together, the results from the current study suggest that the gut dysbiosis associated with *T. gondii* infection in our model may represent a secondary phenomenon but may also precede and facilitate the infection.

The present study successfully induced small bowel inflammation and established its association with the P2X7 pathway. Nevertheless, it also had certain important limitations. First, very few experiments and replicates were performed mainly because of technical difficulties associated with mouse care and the aggressiveness of the *T. gondii* infection. Nonetheless, the results of the study revealed several significant differences between treatment groups and high consistency overall. Second, future investigations should consider using P2X7 antagonists and different animal models and experimental protocols. They should collect samples throughout the study to evaluate the dynamic changes that may occur in any of the parameters measured and particularly those involving the gut microbiota. Finally, antibiotics could be administered in certain treatments to help elucidate the mechanistic roles of the gut microbiota in the model and detect any potential synergism with the P2X7 pathway in activating the inflammasome and conferring host protection against *T. gondii* infection.

## 5. Conclusions

Defective innate immunity in P2X7-deficient mice may render them unable to effectively combat *T. gondii* infection. They mount weaker, less complete inflammatory responses than their WT counterparts. Nonetheless, P2X7-deficient mice infected with *T. gondii* present with wider ileal tissue disruption and more severe consequences than infected WT mice. Therefore, the P2X7 receptor is vital to the host’s defense against peroral *T. gondii* infection. It limits parasite-induced intestinal immunopathology, regulates the systemic immune response, and restrains parasite dissemination in the host. Our findings suggest that the P2X7 pathway also shapes the gut microbiota, which may in turn directly or indirectly determine the outcome of the infection challenge. In P2X7 deficiency, indigenous gut microbiota may create an environment favoring increased susceptibility to *T. gondii* infection. Furthermore, abnormal host-microbial interactions in P2X7 deficiency may foster secondary dysbiosis that exacerbates tissue damage.

## Figures and Tables

**Figure 1 biomedicines-11-00555-f001:**
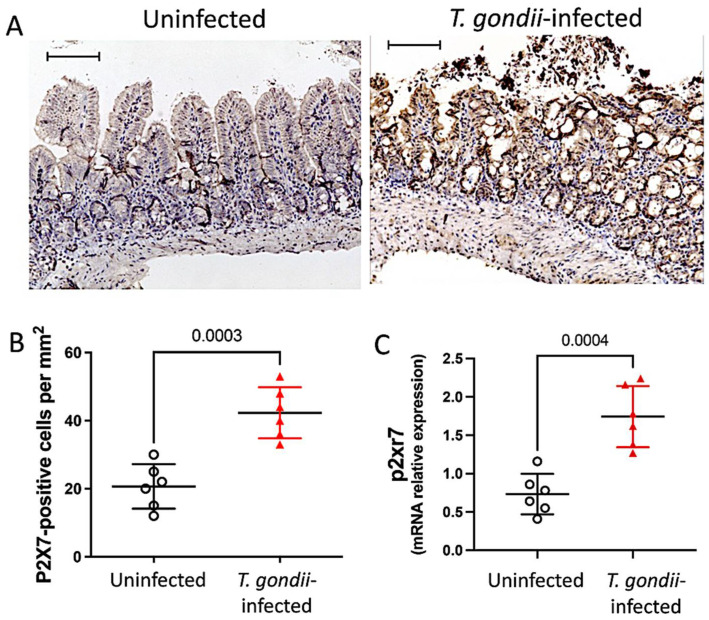
*T. gondii* infection upregulates P2X7 in mouse ileum. Immunohistochemical staining with anti-P2X7 antibody was performed for computerized quantification and analysis of protein distribution in the ileum in the presence and absence of *T. gondii* infection. Bars = 50 µm. Numbers of positive cells/mm^2^ are presented as means ± standard deviation (SD) (**A**,**B**). mRNA levels in ileal samples were measured by quantitative real-time PCR and confirmed that *T. gondii* infection upregulates P2X7 in the ileum. Data are means ± SD (**C**). Analysis by unpaired *t*-test. Significant differences are presented.

**Figure 2 biomedicines-11-00555-f002:**
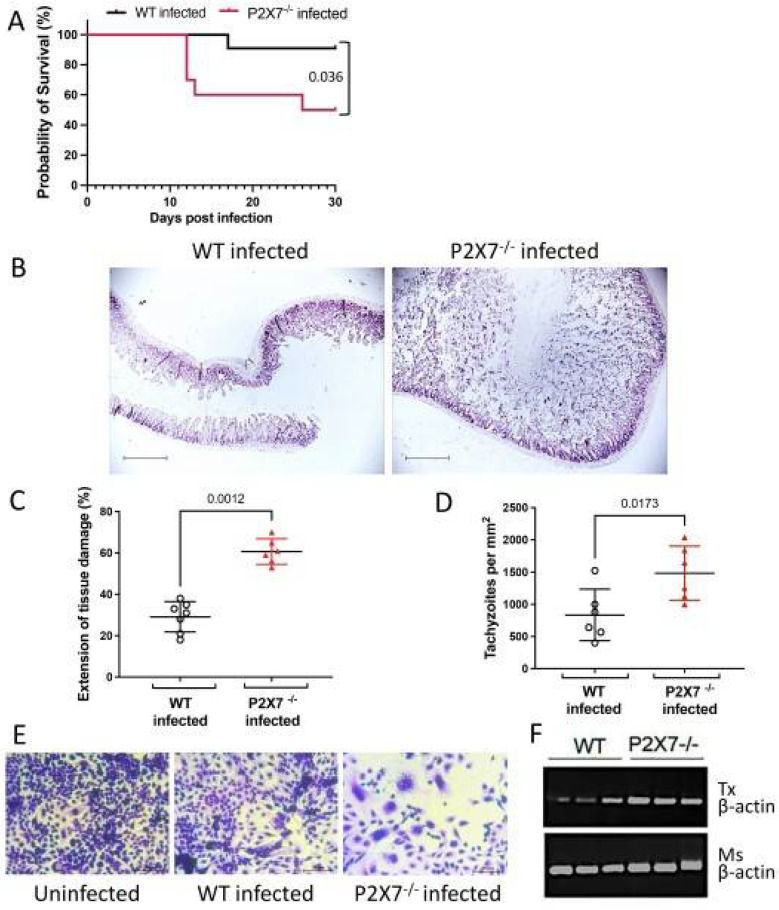
The P2X7 receptor promotes mouse survival by limiting tissue damage and parasite load. The survival curves showed that the infected P2X7−/− mice had significantly higher mortality than the infected WT mice from day 10 onwards (**A**). Data are means ± SD of 11 (WT) and 10 (P2X7−/−) mice/group. Histopathological analysis by hematoxylin and eosin (H&E) staining of the ileum shows more extensive tissue damage in the infected P2X7−/− mice than the infected wild-type (WT) mice. Data are means ± SD of seven (WT) and six (P2X7−/−) mice/group. (**B**,**C**). The number of tachyzoites was higher in the ileal tissue of the infected P2X7−/− mice than the infected WT mice (**D**). Data are means ± SD of six mice/group. ANOVA was performed and followed by the post hoc Tukey’s test. A plaque image of the monolayer showed that the cells treated with the ileal supernatant of infected P2X7−/− mice were more severely damaged than those subjected to the ileal supernatant of infected WT mice. Representative micrograph of three mice/group (**E**). The mRNA levels in the ilea of the infected mice were measured by RT-PCR. *T. gondii* β-actin was more abundant in the infected P2X7−/− mice than the infected WT mice. Amplification products of three mice/group (**F**).

**Figure 3 biomedicines-11-00555-f003:**
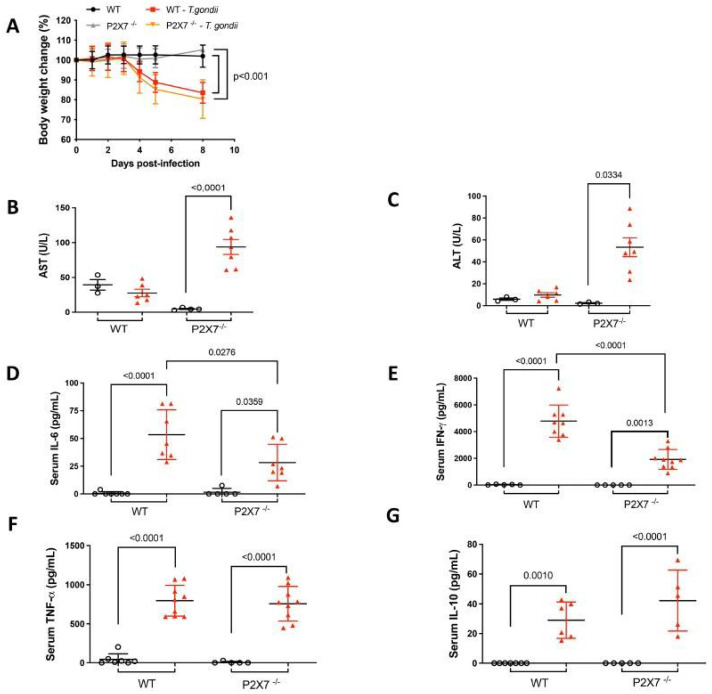
The P2X7 receptor protects against liver damage and inflammatory cytokine production in *T. gondii*-infected mice. Compared to uninfected mice, those with *T. gondii* infection underwent progressive weight loss. Body weight was monitored in 5–10 mice/group and the *p*-value was determined by a log-rank test (**A**). Serum aminotransferases were measured for 3–7 mice/group. There were increased AST (**B**) and ALT (**C**) levels in the P2X7−/− infected mice. Serum IL-6 (**D**), IFN-γ (**E**), TNF-α (**F**), and IL-10 (**G**) levels were measured for 7–9 mice/group and all were elevated in the *T. gondii*-infected animals. However, IL-6 (**D**) and IFN-γ (**E**) increased to a significantly less extent in the infected P2X7−/− mice than in the infected WT mice. Data were subjected to ANOVA and the post hoc Tukey’s test was used for multiple comparisons. Data are means ± standard error of the mean (SEM) and significant differences are shown.

**Figure 4 biomedicines-11-00555-f004:**
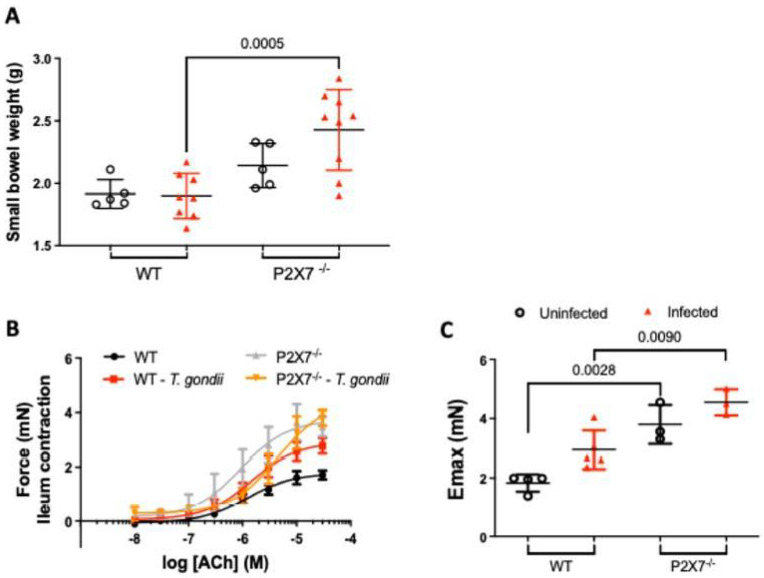
Role of the P2X7 receptor in gut physiology. P2X7 receptor deficiency results in increased intestinal weight in infected mice. Data are means ± standard deviation (SD) of 5–8 mice/group. They were subjected to ANOVA and the post hoc Tukey’s test was used for multiple comparisons. (**A**). Concentration–response curves for contraction induced by ACh (0.01–30 μM) in an ileal segment. Contraction was ACh dose-dependent in the ilea excised from infected wild-type (WT) and P2X7−/− mice as well as uninfected mice. Data are means ± standard error of the mean (SEM) of 3–5 mice/group. They were subjected to ANOVA and the post hoc Tukey’s test was used for multiple comparisons. (**B**). Maximal response (E_max_) to Ach-induced contraction showed that infected P2X7−/− mice had relatively increased ileal contraction. Data are means ± SD of 3–5 mice/group (**C**). Significant differences are shown.

**Figure 5 biomedicines-11-00555-f005:**
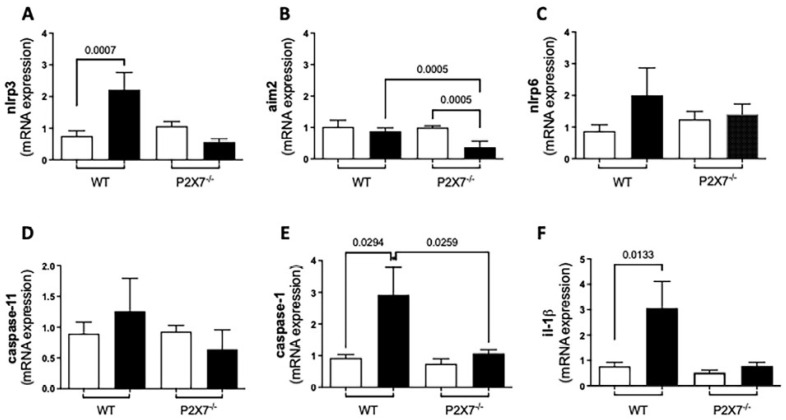
*T. gondii* infection promotes an imbalance in the mRNA expression of inflammasome components measured by quantitative real-time PCR in ileal samples. Significant increases in *NLRP3* (**A**), *Caspase-1* (**E**), and *IL-1β* (**F**) were observed in infected wild-type (WT) mice but not in infected P2X7−/− mice. In contrast, *AIM2* decreased following infection to lower levels in P2X7−/− mice than WT mice (**B**). *NLRP6* (**C**) and *Caspase-11* (**D**) did not significantly change. Data are means ± SEM of 8–13 mice/group. They were analyzed by ANOVA and the post hoc Tukey’s test was used for multiple comparisons. Significant differences are presented.

**Figure 6 biomedicines-11-00555-f006:**
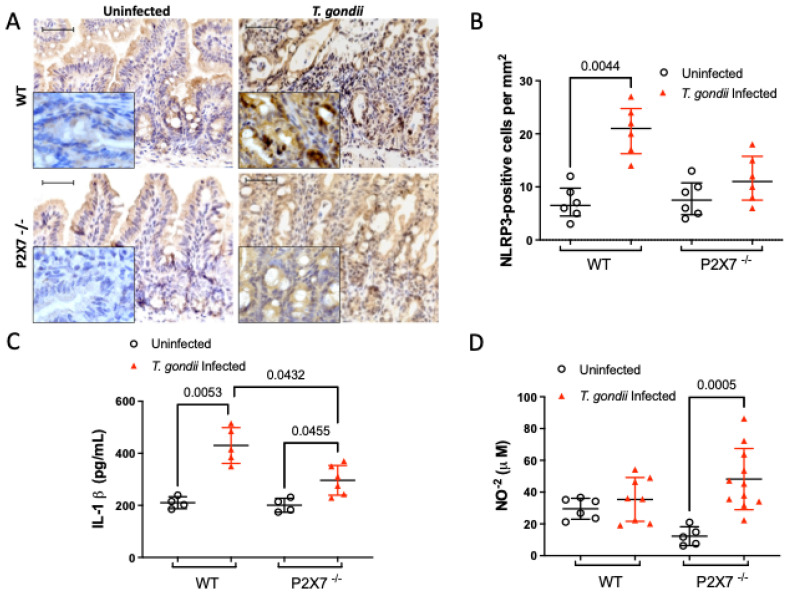
The P2X7 receptor modulates NLRP3 expression and IL-1β and NO production in the ilea of *T. gondii*-infected mice. Immunohistochemical staining showed significant NLRP3 upregulation in infected wild-type (WT) mice but not in infected P2X7−/− mice. Inserts represent magnified micrographs obtained under oil immersion. Data are shown as the means ± standard deviation (SD) of 6–7 mice/group (**A**,**B**). IL-1β was significantly more upregulated in ileal lysates from infected WT mice than it was in those from infected P2X7−/− mice. Data are means ± SD of 4–6 mice/group (**C**). NO production was measured by the Greiss method. NO production was significantly higher in the infected P2X7−/− mice than in the infected WT mice. Data are means ± SD of 5–11 mice/group (**D**). They were subjected to ANOVA and the post hoc Tukey’s test was used for multiple comparisons.

**Figure 7 biomedicines-11-00555-f007:**
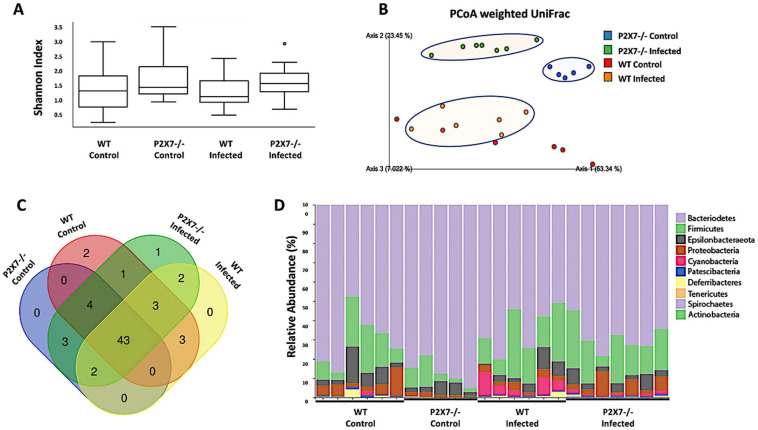
*T. gondii* infection and the P2X7 receptor modulate the intestinal microbiota. Alpha-diversity analysis using the Shannon index showed an upward trend in fecal microbial diversity in the absence of the P2X7 receptor (**A**). β-diversity analysis using weighted UniFrac PCoA clusters P2X7−/− away from WT mice and *T. gondii*-infected mice away from control mice (**B**). Venn diagram showing logical associations among groups (**C**). Differential abundance analysis of taxonomic profiles depicting phylum-level microbial composition (**D**). Sequencing was performed on samples from two independent experiments involving 2–5 mice/group.

## Data Availability

Materials such as protocols, analytical methods, and study material are available upon request to interested researchers. The raw data supporting the conclusions of this manuscript will be made available by the authors without undue reservation to any qualified researcher. Submission ID: SUB12533790; BioProject: ID: PRJNA925189.

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
