# Peer review of "P2X7 Receptor Modulation of the Gut Microbiota and the Inflammasome Determines the Severity of *Toxoplasma gondii*-Induced Ileitis"

_biomedicines, 2023, doi:10.3390/biomedicines11020555_

Round 1

Reviewer 1 Report

The manuscript provides a valuable contribution to the understanding of the role of the P2X7 receptor in in intestinal infections. It demonstrates that P2X7 is involved - via activation of distinct cytokines - in the control of intestinal parasites.

Addressing some of the minor points mentioned below would further improve the clarity of the work.

1.) L83: “Thus, we hypothesized that compositional changes in the gut microbiome may precede the onset of T. gondii-induced ileitis.”

Couldn't it also be that the changes in the microbiome are a result of the ileitis? This should be discussed.

2.) L211: The difference in tissue alteration measured as shown in Fig. 2 ("The T. gondii infection damaged the ileal mucosa to a significantly greater extent in the P2X7-/- than in the WT mice.") and Suppl. Fig. 1 ("T. gondii infection induces severe ileal inflammation and damage. Histopathological analysis by hematoxylin and eosin (HE) staining of the ileum shows severe inflammatory changes and tissue damage initiated by T. gondii-infection, but apparently not affected by the expression of the P2X7R (A, B).") are not clear for me. This could be better explained.

3.) L242: “Data are means ± SD of six mice/group.”

What does this statement belong to? If A represents 6 mice per group, then the steps are too small.

4.) L246: “Data are means ± SD of six mice/group. ANOVA was performed and followed by the post hoc Tukey’s test.”

This statement belongs to C+D? In C (wt), I see 7 data points.

5.) L276: “The P2X7 receptor protects against damage …”

Unclear statement. Which damage? Liver?

6.) L294: “However, there was a significant difference in Emax for ACh-induced contraction between the infected WT mice (1.79 ± 0.12 mN, n = 4) and the infected P2X7-/- mice (3.71 ± 0.38 mN, n = 3, p = 0.01).”

This is the difference between the uninfected WT mice and the uninfected P2X7-/- mice.

7.) L299: “This increase was relatively 299 greater …”

There are no statistics given for the greater relative increase.

8.) L324: “Significant increases in … Caspase-1 (E) … were observed in infected wild-type (WT) mice but not in infected P2X7-/- mice.”

This is apparently in contrast to Suppl. Figure 4 ("caspase-1, … were significantly increased in infected mice, but no difference was detected regarding the expression of the P2X7R"). Please discuss.

9.) L354: “control P2X7R-/-: 43,565 ± 935”

Is that the right SD?

10.) Fig 7A: What is the meaning of the asterisk?

11.) L460: “Nevertheless, we hypothesize that P2X7 deficiency may reduce ATP release from epithelial and enterochromaffin cells which, in turn, lowers inhibitory junctional potentials and, by extension, potentiates the contractions induced by ACh. Under normal conditions, neuronal P2X7 receptors have relatively less control of tonic inhibition on excitatory cholinergic motility [31]. For this reason, Ach-induced contraction was comparatively stronger in the ilea of the infected P2X7-/- mice than it was in those of the infected WT mice.”

It should be considered, that in the experiments, ACh acts directly on the smooth muscle cells and neurons are not involved.

12.) Supplementary figures S6 to S9: Ordinate labels and asterisks are hardly recognizable. What do the significance levels refer to?

13.) Supplementary Figure S8: “reductions in Prevotellaceae were observed in WT and P2X7-/- mice”

I do not see a reduction in Prevotellaceae in wt.

Author Response

Reviewer #1

The manuscript provides a valuable contribution to the understanding of the role of the P2X7 receptor in in intestinal infections. It demonstrates that P2X7 is involved - via activation of distinct cytokines - in the control of intestinal parasites.

R: We thank this reviewer for his/her interest in our study and for giving us the opportunity to improve the quality of the manuscript.

Addressing some of the minor points mentioned below would further improve the clarity of the work.

1.) L83: “Thus, we hypothesized that compositional changes in the gut microbiome may precede the onset of T. gondii-induced ileitis.”

Couldn't it also be that the changes in the microbiome are a result of the ileitis? This should be discussed.

R: We understand this reviewer’s concern and agree with this comment. To summarize, yes, the changes in the microbiome could be primary (genetic defect- due to P2X7 deficiency) or secondary (due to T. gondii infection and inflammation). We actually have some evidence from the literature for both possibilities. Therefore, we understand that we must elaborate more on this issue raised by this reviewer in the discussion section. We added explanatory text in the Discussion section to the 8th, 9th, and 10th paragraphs.

2.) L211: The difference in tissue alteration measured as shown in Fig. 2 ("The T. gondii infection damaged the ileal mucosa to a significantly greater extent in the P2X7-/- than in the WT mice.") and Suppl. Fig. 1 ("T. gondii infection induces severe ileal inflammation and damage. Histopathological analysis by hematoxylin and eosin (HE) staining of the ileum shows severe inflammatory changes and tissue damage initiated by T. gondii-infection, but apparently not affected by the expression of the P2X7R (A, B).") are not clear for me. This could be better explained.

R: We agree with this comment, and we thank this reviewer for the opportunity to improve the clarity of text regarding this crucial point in the manuscript. In fact, what we meant was showing that the tissue damage and inflammatory reaction due to the infection had a far greater extent in the P2X7-/- deficient mice, with several centimeters of the small bowel affected towards the jejunum. In contrast, the WT animals developed severe inflammation only in the terminal ileum. However, when analyzing the terminal ileum alone, as we did for most experiments, the groups showed no significant difference.

3.) L242: “Data are means ± SD of six mice/group.”

What does this statement belong to? If A represents 6 mice per group, then the steps are too small.

R: We understand this reviewer’s doubt or concern, and we agree we made a mistake. The legend is wrong. We analyzed 11 (WT) and 10 (P2X7-/-) mice, respectively.

4.) L246: “Data are means ± SD of six mice/group. ANOVA was performed and followed by the post hoc Tukey’s test.”

This statement belongs to C+D? In C (wt), I see 7 data points.

R: We apologize again for the mistakes and misunderstandings regarding the numbers in each experiment or figure. We had too many experiments, with different numbers in each. Corrections were performed according to this reviewer’s indication.

5.) L276: “The P2X7 receptor protects against damage …”

Unclear statement. Which damage? Liver?

R: We understand this reviewer’s concern, and we agree to add “liver damage” in the sentence for clarity.

6.) L294: “However, there was a significant difference in Emax for ACh-induced contraction between the infected WT mice (1.79 ± 0.12 mN, n = 4) and the infected P2X7-/- mice (3.71 ± 0.38 mN, n = 3, p = 0.01).”

This is the difference between the uninfected WT mice and the uninfected P2X7-/- mice.

R: We agree with this comment, as our text was not clear in that part of the manuscript. Additional explanatory text was added to the manuscript. In fact, ACh induced stronger ileal contraction in the uninfected P2X7-/- mice than in the uninfected WT mice (Figure 4 C). Moreover, the infected WT mice presented with stronger ileal contractions (2.91 ± 0.22 mN, n = 5) than the uninfected WT mice (4.39 ± 0.46 mN, n = 3, p = 0.009) (Figure 4 C).

7.) L299: “This increase was relatively 299 greater …”

There are no statistics given for the greater relative increase.

R: We apologize for the mistake and made appropriate corrections according to this reviewer’s indication. We added the p-value (p=0.009)

8.) L324: “Significant increases in … Caspase-1 (E) … were observed in infected wild-type (WT) mice but not in infected P2X7-/- mice.”

This is apparently in contrast to Suppl. Figure 4 ("caspase-1, … were significantly increased in infected mice, but no difference was detected regarding the expression of the P2X7R"). Please discuss.

R: We understand this reviewer’s concern and added some additional text to the discussion, as suggested. In our study, although the protein levels of caspase-1 are increased in both WT and P2X7-/- mice following T. gondii infection, the respective caspase-1 mRNA increased only in the WT group. This could be explained by the existence of additional pathways for activating caspase-1, such as the direct induction by microorganisms, for example. In a study by Franchi et al, intracellular Salmonella or Listeria were shown to activate pro-caspase-1 in macrophages, even in the context of P2X7 receptor deficiency (Journal of Biological Chemistry, 2007). Therefore, we speculate that available cytosolic pro-caspase-1 may be activated, at least in part, by alternative pathways, independent of P2X7, and added a new reference (#34, Franchi et al, 2007).

9.) L354: “control P2X7R-/-: 43,565 ± 935”

Is that the right SD?

R: Yes this SD is correct. Some confusion may have been caused by the commas applied to thousands and millions. Thousands of reads were obtained following the sequencing experiment. Therefore, commas were used to separate groups of thousands.

10.) Fig 7A: What is the meaning of the asterisk?

R: That was a misunderstanding. Actually, the mark is a dot, representing an outlier (and not an asterisk).

11.) L460: “Nevertheless, we hypothesize that P2X7 deficiency may reduce ATP release from epithelial and enterochromaffin cells which, in turn, lowers inhibitory junctional potentials and, by extension, potentiates the contractions induced by ACh. Under normal conditions, neuronal P2X7 receptors have relatively less control of tonic inhibition on excitatory cholinergic motility [31]. For this reason, Ach-induced contraction was comparatively stronger in the ilea of the infected P2X7-/- mice than it was in those of the infected WT mice.”

It should be considered, that in the experiments, ACh acts directly on the smooth muscle cells and neurons are not involved.

R: We agree with this comment, and we added more explanatory text in legends and figures.

We performed concentration-response curves to acetylcholine which mediated myogenic (longitudinal) contractions. However, the remaining post-synaptic neurons in the preparation could still release NANC neurotransmitters such as ATP, which in that case, could affect the ACh-mediated contraction. P2X7 receptors are also located in myenteric neurons. In rat chemical-induced colitis model, the P2X7 antagonist A804598 induced a significant enhancement of the contractions, which seemed to depend on the NO pathway, since the neuronal nitric oxide synthase inhibitor (NPA) blunted the A804598 effects. Finally, P2X7R stimulation with BzATP induced a marked reduction of colonic contraction under inflammation (Antonioli et al., 2014, doi:10.1371/journal.pone.0116253). Therefore, the lack of a P2X7 receptor could partially explain the enhanced ileum contractions during Toxoplasma infection.

12.) Supplementary figures S6 to S9: Ordinate labels and asterisks are hardly recognizable. What do the significance levels refer to?

R: We agree with this comment, and we included new comparative graphs (in the supplementary contents), showing exactly the relevant differences between the groups. Consequently, we had to update some parts of the text (results) and legends.

13.) Supplementary Figure S8: “reductions in Prevotellaceae were observed in WT and P2X7-/- mice”

I do not see a reduction in Prevotellaceae in wt.

R: We agree with this comment, and we provided an appropriate correction.

Thank you!

Reviewer 2 Report

In this paper the authors provide a description of the impact of P2X7 receptor in the inflammatory response caused by a model of ileitis induced by Toxoplasma Gondii. 

Below are my comments.

The authors need to standardize the use of P2X7 ko or P2X7R ko. Established that are the same thing it is confusing if the extension is differently used in the text and in the legend for example.

The signals in the immunohistochemical images for PTX7 receptor (Fig.1) and NLPR3 (Fig.6) are not so clear and the lower resolution provided do not allow to understand the differences. I suggest providing an inset with higher magnification to focus on the specific antibody signal. 

In this paper DOI: 10.3390/ijms23094616 the induction of the same mechanisms are described in colorectal cancer. Do the authors can comment on this?

The authors cite an effect in liver inflammation reporting variations in AST and ALT levels. Given the importance of the role of microbiota composition in the liver diseases (gut-liver axis), do the authors have the chance to measure the composition in terms of SCFAs metabolites or FXR pathway? They are encouraged to provide a comment on this aspect. Do the authors have data on parasitic load in the liver?

In intestinal inflammation the role of Th17 cells is still a matter of debate. Do the authors can provide some data regarding this important population? In particular I the authors can perform FACS analysis on intraepithelial lymphocytes at baseline and upon T.Gondii infection?

Do the authors know form literature if the specific taxa described in the P2X7 mice microbiota are associated with modulation in immune cell distribution?

Since the weight loss remain stable until 20 days, I ask if the sacrifice can be postponed to also analyze better the adaptive immune response?

Author Response

Reviewer #2

In this paper the authors provide a description of the impact of P2X7 receptor in the inflammatory response caused by a model of ileitis induced by Toxoplasma Gondii. 

Below are my comments.

  1. The authors need to standardize the use of P2X7 ko or P2X7R ko. Established that are the same thing it is confusing if the extension is differently used in the text and in the legend for example.

R: We agree with this comment, and we apologize for the mistake. Appropriate corrections were made, removing the extra R whenever present in the text, legends, and Figures. Therefore, we needed to update Figures 1, 6, and 7.

  1. The signals in the immunohistochemical images for PTX7 receptor (Fig.1) and NLPR3 (Fig.6) are not so clear and the lower resolution provided do not allow to understand the differences. I suggest providing an inset with higher magnification to focus on the specific antibody signal. 

R: We understand this reviewer’s point of view, and we made new image captures, preparing a new panel containing the inserts, as suggested. We prepared a new Figure 6, updating the corresponding legend.

  1. In this paper DOI: 3390/ijms23094616the induction of the same mechanisms are described in colorectal cancer. Do the authors can comment on this?

R: We thank this reviewer for his/her attentive reading of our manuscript. The paper mentioned by the reviewer belongs to a long list of studies from our group regarding P2X7 purinergic receptors. Most papers from our group focus on intestinal diseases, with human studies in IBD, in vitro studies, different experimental models of colitis, colorectal cancer, and ileitis, and frequently using P2X7 knockout mice, and different P2X7 inhibitors. The mechanisms involved in mucosal inflammation and also in P2X7 functionality are considerably conserved and repetitive. Therefore, we always need to study morphologic and cellular components within the tissue, from epithelial to lamina propria cells, counting cells, analyzing epithelial damage, apoptosis, autophagy, necrosis, fibrogenesis, measuring the expression of chemokines and cytokines, myeloperoxidase, oxidative stress, activation of intracellular signaling pathways and the inflammasome. Occasionally, we have added to our techniques, video-colonoscopy follow-up, endoluminal ultrasound, electrophysiological measurements, and analysis of the gut microbiota. The paper on the role of P2X7 in the development of CA-CRC has been as added to the Discussion section (new reference #50).

  1. The authors cite an effect in liver inflammation reporting variations in AST and ALT levels. Given the importance of the role of microbiota composition in the liver diseases (gut-liver axis), do the authors have the chance to measure the composition in terms of SCFAs metabolites or FXR pathway? They are encouraged to provide a comment on this aspect. Do the authors have data on parasitic load in the liver?

R: We appreciate this reviewer’s pertinent comment. However, unfortunately, we did not have a chance to analyze other pathways or production of SCFAs. In fact, the study was already too long, and we needed to leave most of it as supplementary content, to follow the journal instructions. Nevertheless, in a previous study from our group, we showed that the liver of infected P2X7-/- mice had smaller granulomas, but increased parasite load per granuloma (DOI: 10.1016/j.imbio.2016.12.007). This was added to the text in the Discussion section (keeping the same reference, #24).

  1. In intestinal inflammation the role of Th17 cells is still a matter of debate. Do the authors can provide some data regarding this important population? In particular I the authors can perform FACS analysis on intraepithelial lymphocytes at baseline and upon T.Gondii infection?
  2. Do the authors know form literature if the specific taxa described in the P2X7 mice microbiota are associated with modulation in immune cell distribution?

R: This is the second study from our group analyzing the fecal microbiota in P2X7 deficient mice. The first one, to our knowledge, was exactly the one mentioned above, from our group, working with colitis-associated cancer. Therefore, we included a reference from our group (#50), and added text to the Discussion section.

  1. Since the weight loss remain stable until 20 days, I ask if the sacrifice can be postponed to also analyze better the adaptive immune response?

R: Actually, in the model used for inducing ileitis, the body weight change starts to be evident after day 4 (Fig. 3A). Deaths, however, started after day 10 (Fig. 2A). Due to difficulties with the model, and the previous experience, with high mortality rates after day 7, we chose to sacrifice on day 8.

Thank you!

Round 2

Reviewer 2 Report

The authors fully address my questions except for point 5 in which I cannot see the answer but I think is only an oversight.